# *Drosophila* RhoGAP18B regulates actin cytoskeleton during border cell migration

**Fengyun Lei[1], Xiaoqing Xu[1], Jianhua Huang[1], Dan Su[2]\*, Ping Wan[1]\***

**1** Laboratory of Molecular Biology, School of Life Science, Jiangxi Science and Technology Normal University, Nanchang, China, **2** Key Laboratory of Animal Model of TCM Syndromes of Depression, Jiangxi Administration of traditional Chinese Medicine, Jiangxi University of Chinese Medicine, Nanchang, China

\* pingwan2012@hotmail.com (PW); sud94@aliyun.com (DS)

**Data Availability Statement:** All relevant data are within the paper and its Supporting Information files.

**Funding:** This work was supported by the National Natural Science Foundation of China (31660330 for

## Abstract

*Drosophila* RhoGAP18B was identified as a negative regulator of small GTPase in the behavioral response to ethanol. However, the effect of RhoGAP18B on cell migration is unknown. Here, we report that RhoGAP18B regulates the migration of border cells in *Drosophila* ovary. The *RhoGAP18B* gene produces four transcripts and encodes three translation isoforms. We use different RNAi lines to knockdown each RhoGAP18B isoform, and find that knockdown of RhoGAP18B-PA, but not PC or PD isoform, blocks border cell migration. Knockdown of RhoGAP18B-PA disrupts the asymmetric distribution of F-actin in border cell cluster and increases F-actin level. Furthermore, RhoGAP18B-PA may act on Rac to regulate F-actin organization. Our data indicate that RhoGAP18B shows isoform-specific regulation of border cell migration.

## 1. Introduction

Guided cell migration, or chemotaxis, plays a key role in many physiology and pathological events, such as vertebrate neural crest migration, and tumor metastasis [1–3]. Border cell migration in *Drosophila* egg chambers is a well characterized genetic model to study the mechanisms of chemotaxis or collective cell migration. *Drosophila* ovaries consist of multiple egg chambers in different developmental stages. Each egg chamber, which is covered by a monolayer of epithelial follicle cells, contains an oocyte and fifteen nurse cells. Border cells are a group of 6–10 cells that initiate from the follicle cells at the anterior tip of the egg chamber. During stage 9, border cells migrate through the nurse cells and reach the border of oocyte by stage 10, approximately 6 h after the initiation of migration. After reaching the oocyte, the border cell cluster will assist in forming a channel for sperm to enter oocyte during fertilization. During migration, the border cell cluster exhibits a tumor-like invasive manner, producing a forward-directed protrusion and moving as a coherent cluster collectively. Therefore, border cell migration is an ideal model to study collective cell migration of a group of cells [4, 5].

To start a migration, most cells need to extend their plasma membrane protrusions, known as lamellipodia, which are organized with actin filament in a guided direction. During migratory process, the dynamic assembly and disassembly of actin filaments are precisely regulated to complete a guided cell migration. Rac is among the key regulators of actin dynamics that

P.W. and 81860702 for D.S.). https://www.nsfc.gov.cn/.

**Competing interests:** The authors have declared that no competing interests exist.

drives the plasma membrane extension in lamellipodia [6]. As a Rho GTPase, Rac plays actin-dependent function like a common molecular switch. Rac function must be precisely controlled during border cell migration, as expressing a dominant-negative form of Rac (RacN17) and expressing a constitutive active form of Rac (RacV12) both results in severe defects of border cell migration [7–9]. Local change of Rac activity results in changes on collective morphology and affects border cells to extent directed protrusion [10–12]. Moreover, expressing RacV12 can induce elevated levels of F-actin in border cells [13]. Rac signals downstream to Lim kinase (LimK), which can mediate the activity of actin dynamic promoting factors [14]. The Rho GTPase activity of Rac is regulated by the GTPase cycle between an inactive guanosine diphosphate (GDP) form and an active guanosine triphosphate (GTP) form. The GTPase activity is promoted by guanine nucleotide exchange factors (RhoGEFs), which facilitate GTP loading and binding of downstream effector, and inactivated by Rho GTPase-activating proteins (RhoGAPs), which stimulate GTP hydrolysis [15]. Therefore, both RhoGEFs and RhoGAPs are critical for Rho GTPase activity. However, their activities in regulating actin cytoskeleton and cell migration have not been well elucidated.

The *Drosophila RhoGAP18B* locus encodes three protein isoforms, PA, PC and PD, which differ extensively in their N termini and share a conserved GAP domain in their C termini. Distinct RhoGAP18B isoforms mediate different behavioral response of fly to ethanol: PC regulates ethanol-induced sedation, while PA regulates ethanol-induced hyperactivity [16]. Different RhoGAP18B isoforms also regulate actin cytoskeleton by acting on distinct Rho GTPases in cultured *Drosophila* Schneider (S2) cells [17]. Rho GTPase mediated actin dynamics plays a key role during border cell migration in *Drosophila* ovary [7, 18]. Here, we investigated the function of RhoGAP18B isoforms in *Drosophila* border cell migration. We find that PA isoform of RhoGAP18B has a modulator role in the regulation of border cell migration, while PC and PD are dispensable for this process. We also demonstrate that RhoGAP18B-PA mediates F-actin organization in border cell cluster probably through Rac. Thus, our data shows RhoGAP18B has isoform-specific regulation of border cell migration.

## 2. Methods

### 2.1 *Drosophila* genetics

All fly stocks involved in the experiment were obtained from the Bloomington *Drosophila* Stock Center (BDSC), except PA-RNAi (7531R-2) from National Institute of Genetics Stock Center (NIG-FLY), PC-RNAi (28047) and PD-RNAi (104589) from Vienna Drosophila Resource Center (VDRC). Most *Drosophila* stocks were maintained and crossed at 25˚C according to standard procedures. For *slbo-Gal4* and *c306-Gal4* crosses, flies were cultured at 29˚C for 3 days before ovary dissection to produce optimal GAL4/UAS transgene expression. The UAS-PA clone was generated by Quanyang Biotechnology (Suzhou) Co., Ltd. The full-length PA coding sequence, with two EcoRI sites added in the 5' and 3' ends (S1 Fig), was synthesized and subcloned to the pUAS-attB plasmid with EcoRI site. The clone was sequence-verified. The transgenic line was established by Core Facility of Drosophila Resource and Technology, CEMCS, CAS. The attP2 site was used at 68A4.

### 2.2 Immunostaining and microscopy

Ovary dissection was carried out in phosphate-buffered saline (PBS) and then fixed in formaldehyde solution and PBS mixture (4% formaldehyde) for 10 min. After three washes in PBS, ovaries were washed in PBT for 10 min and then stained with reagents in PBT for 30 min. Reagents were as follows: Rhodamine phalloidin (1:100, Sigma), DAPI (1:10, Sangon Biotech). After three washes in PBT, ovaries were washed in PBS for 10 min to deflate bubbles. Dissected

egg chambers were mounted on slides in 60% glycerol prior to imaging. Confocal images were obtained using a Leica TCS SP8 or an Olympus FV1000 confocal microscope. Images were processed by ImageJ, GraphPad Prism software and Photoshop.

## 2.3 RT-qPCR

Total RNA was extracted from adult fly or adult fly ovary using RNAiso Plus reagents (9108, TaKaRa) according to the manufacturer's protocol. One microgram of total RNA from each specimen was reverse-transcribed to cDNA using a PrimeScript™ RT reagent Kit with gDNA Eraser (RR047A, TaKaRa). Quantitative real-time reverse transcription PCR (RT-qPCR) was performed with a RT-qPCR instrument (Biorad) using TB Green® Premix Ex Taq™ II (RR820A, TaKaRa), 0.4 μM of forward and reverse primers, and 2 mL diluted cDNA in a total volume of 20 mL reaction system. Primers for *RA/RB*, *RC*, *RC/RD* and the internal control *β-actin* were as follows: *RA/RB* forward 5'-ATCATCCACCGCATCCTTGTC-3' and reverse 5'-ATGGCGATGGTTGGAGGTCAG-3'; *RC* forward 5'-ACAACAGCGACAACAACCACC-3' and reverse 5'-GGCGAAGGAGAAGGCGAAGAG-3'; *RC/RD* forward 5'-AGATTGGGTCG TGTTGAAGCAG-3' and reverse 5'-CACCACCAGACAGTTGGAGTAGTAGA-3'; *β-actin* forward 5'- GAGCCACCGATCCAGACAGAG-3' and reverse 5'- GCATCCACGAGACCACC-TACA-3'. The PCR program used was 95°C for 30 seconds, followed by 40 cycles of 95°C for 5 seconds and 60°C for 34 seconds. The relative amount of target gene expression in each specimen was normalized to *β-actin*.

## 2.4 Quantification of fluorescence intensity

The quantification methods are similar to that described previously [13]. For measurement of the front/back ratios, each border cell cluster was divided into two halves along the migration direction. Fluorescence intensity (FI) and area were measured in ImageJ software (NIH) for each half region; the front/back ratios were calculated as [front half FI/front half area] divided by [back half FI/back half area]. For measurement of F-actin's levels in border cells, membrane F-actin of nurse cells adjacent to border cell cluster was used for normalization of fluorescence. The normalized FI was determined as [border cells FI/ border cells area] divided by [nurse cells membrane FI/nurse cells membrane area].

## 3. Results

### 3.1 The PA isoform of RhoGAP18B shows a higher level than PC and PD in *Drosophila* ovary

Data from Flybase and previous studies demonstrate that *RhoGAP18B* gene locus encodes four transcripts: *RA*, *RB* which lacks 519 bases in the 5'UTR of *RA*, *RC* and *RD*, and in turn translates to three protein isoforms: PA, PC and PD [16, 17]. Note that: RA corresponds to RF on FlyBase; RB, RC and RD correspond to the same named transcripts on FlyBase; RH on Flybase was not detected in our and previous studies. The three protein isoforms contain the same conserved GAP domain in their C termini, but differ extensively in their N termini (Fig 1A). First, we used real-time quantitative RT-PCR analysis (qRT-PCR) specific for *RA/RB*, *RC* or *RC/RD* transcripts to identify the expression of these isoforms in the whole body and the ovary of *Drosophila*. In the whole adult fly, *RA/RB*, *RC* or *RC/RD* transcripts showed a similar level. But in the ovary, the amount of *RA/RB* transcripts was much higher than that of *RC* or *RC/RD* (Fig 1B), implying that RhoGAP18B-PA may play a more important role than PC and PD in *Drosophila* ovary.

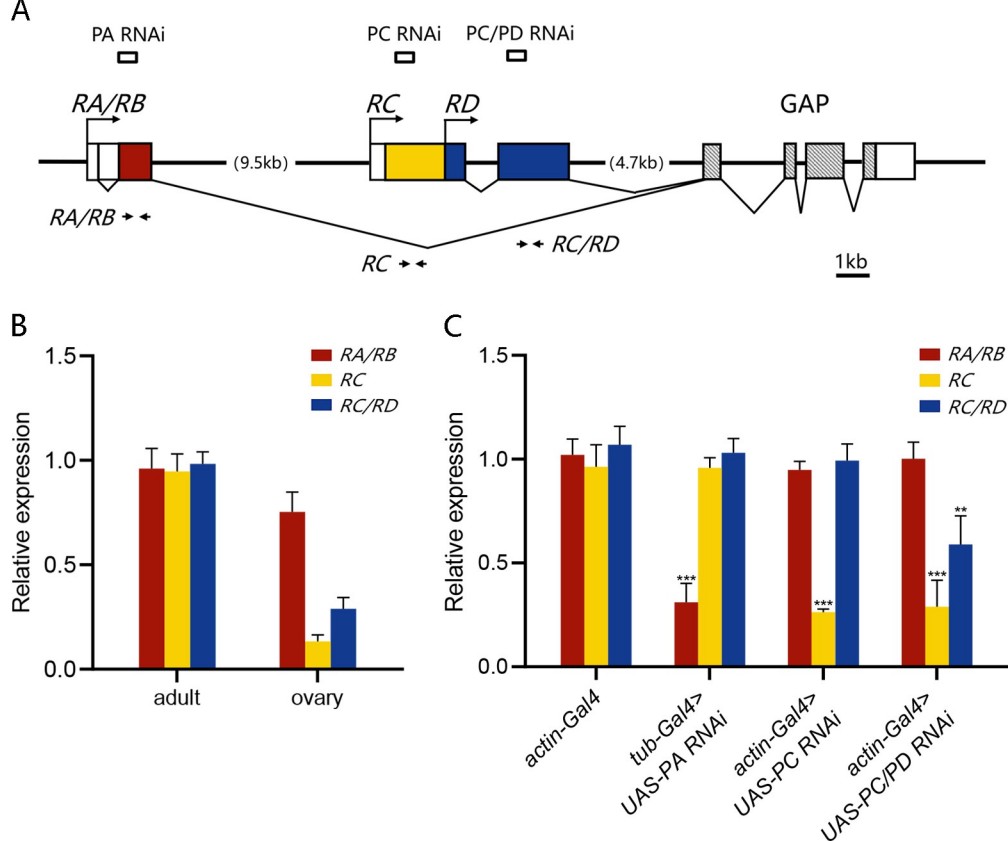

**Fig 1. The RhoGAP18B gene encodes three protein isoforms.** (A). Schematic diagram of the *RhoGAP18B* gene, encompassing about 31 kb of genomic DNA. Exons are shown as boxes and introns as lines. The transcription start sites for the *RA/RB*, *RC*, and *RD* are indicated by crease and arrows. The *RB* transcript is fully contained within *RA*. The three coding regions/proteins are indicated in color at their N-termini. All three isoforms share a common GAP domain at their C-termini, which is shown by stripes. The PD isoform is fully contained within PC. *The targeting sites of three RNAi lines used in experiments* are shown above the gene structure by small boxes. PA RNAi targets to the transcripts *RA/RB* encoding of PA, PC RNAi only targets to the transcripts *RC* encoding of PC, and PC/PD RNAi targets to the transcripts *RC/RD* encoding of PC and PD. RT-PCR primers used in (B and C) are indicated with arrows below the gene structure. Note that: RA corresponds to RF on FlyBase; RB, RC and RD correspond to the same named transcripts on FlyBase. (B) Relative mRNA levels of *RA/RB*, *RC* and *RC/RD* transcripts from wildtype adult fly body and ovary. Transcripts for three protein isoforms show similar expression level in adult fly, but transcipts for PA shows a relatively higher level than that for PC or PD in ovary. (C) Relative mRNA levels of *RhoGAP18B RA/RB*, *RC* or *RD* in fly whole body for control and RNAi flies. PA and PC RNAi knock down corresponding transcripts efficiently (PA, ***P<0.001, t = 11.9, t test, n = 4. PC, ***P<0.01, t = 12.9, t test, n = 4). PC/PD RNAi knocks down *RC/RD* transcripts efficiently (PC, ***P<0.001, t = 8.1, t test, n = 4. PD, **P<0.01, t = 5.9, t test, n = 4). Results are presented as mean ± standard error of mean.

In order to knock down specific isoforms of RhoGAP18B in border cells, we first obtained three fly lines expressing UAS-RNAi which targets distinct region of *RhoGAP18B* gene. PA RNAi targets to the transcripts *RA/RB* encoding of PA, PC RNAi only targets to the transcripts *RC* encoding of PC, and PC/PD RNAi targets to the transcripts *RC/RD* encoding of PC and PD (Fig 1A). The ability of these RNAi lines to knock down *Rho-GAP18B* transcripts was validated by qRT-PCR. We drove the expression of each RNAi using Gal4 expressed ubiquitously, and examined the transcript levels of the three isoforms in whole fly body. PA and PC RNAi significantly reduced the levels of corresponding transcripts to less than 30%. PC/PD RNAi reduced PC and PD transcripts to 30% and 55%, respectively (Fig 1C).

## 3.2 RhoGAP18B-PA is required for border cell migration

The *slbo-Gal4* driver is expressed in border cells and some follicular cells at the anterior side of stage10 oocytes. To find out the roles of RhoGAP18B isoforms in border cell migration, we expressed RhoGAP18B RNAi under the regulation of *slbo-Gal4*. As previously described [19], *slbo-Gal4>UAS-GFP* was used to visualize border cell migration. The migration degree was categorized as 0% (no migration), 25%, 50%, 75%, or 100% (reaching the border) for quantitative analysis in stage 10 egg chambers. In controls, 97% of the border cells completed migration to the anterior border of the oocyte by stage 10 of oogenesis (Fig 2A–2B' and 2F). When PC or PC/PD RNAi was expressed in border cells with *slbo-Gal4*, 95% or 96% of the border cells completed migration respectively, showing no significant changes compared with the control group. In contrast, expression of PA RNAi *resulted in a delayed migration phenotype with only 12%* of border cells completed their migration at stage 10 (Fig 2C, 2C' and 2F). We also drove expression of RhoGAP18B RNAi using *c306-Gal4*, an early anterior follicle cell driver highly expressed in border cells. Similarly, expression of PA RNAi driven by *c306-Gal4* disrupted border cell migration, but expression of PB or PC/PD RNAi did not affect border cell migration (S2 Fig). These experiments suggested that only *RhoGAP18B-PA* was essential for the normal migration of border cells.

To exclude the off-target effect of PA RNAi, we generated UAS-PA transgenic fly and performed rescue experiment to verify whether expression of PA could recover the phenotype caused by PA RNAi. Border cells expressing both PA RNAi and PA finished migration in 85% of stage 10 egg chambers, compared with 12% or 13% in those that expressed PA RNAi alone or PA alone respectively (Fig 2C–2F). The results indicated that the less efficient migration of PA RNAi border cells was indeed due to the lack of PA. The same rescue experiments using *c306-Gal4* also showed that PA rescued the border cell migration defect caused by PA RNAi (S2 Fig). Moreover, we found that overexpression of PA also disrupted border cell migration (Figs 2D, 2D' and 2F and S2). So, these results demonstrated that border cell migration required a precise level of RhoGAP18B-PA.

## 3.3 RhoGAP18B-PA regulates F-actin organization in border cells

RhoGAP18B isoforms are involved in the regulation of actin dynamics in cultured *Drosophila* Schneider (S2) cells [17]. Since a prominent feature of guided migration is the presence of protrusions that is rich in F-actin, we next investigated whether loss of *RhoGAP18B-PA* affects border cell cluster protrusions. We visualized the protrusions with F-actin stained by phalloidin. During migration, the border cell cluster typically forms one or two large protrusions that extend or retract from the leading edge of the cluster as it migrates [20]. In agreement with this, control border cells cluster typically had one or two main protrusions at the front edge (Fig 3A and 3A'). Conversely, when PA RNAi was expressed in border cells the clusters rarely extend protrusions from their front (Fig 3B and 3B'). This result suggested that the border cell clusters failed to form a overall front-back polarity. We calculated the front/back ratio to quantify the front-back polarity of F-actin staining in the border cell clusters and found the polarity was strikingly reduced in PA RNAi expressing *border cell clusters* (Fig 3D). Furthermore, the clusters with *PA RNAi expressing* showed higher F-actin levels than the controls (Fig 3A–3B'). Consistently, quantification results also demonstrate that the F-actin staining was increased in *PA RNAi expressing border cell clusters* (Fig 3E). Moreover, PA overexpression also affected the protrusion forming of border cells (Fig 3C and 3C'). Border cell clusters with PA overexpression showed reduced polarity of F-actin distribution and normal level of F-actin compared with the controls (Fig 3D and 3E). Together, these data indicated that precise level of

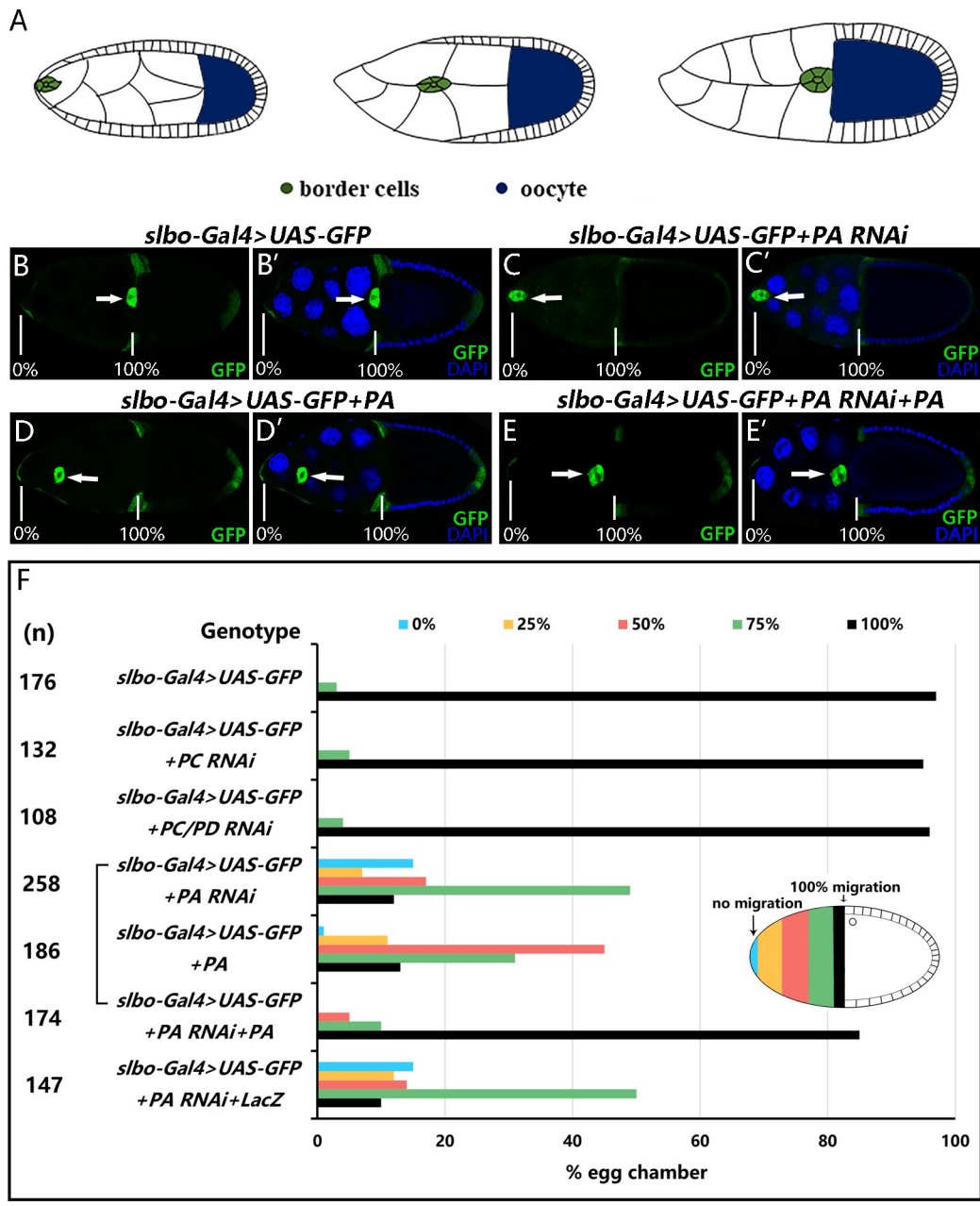

**Fig 2. RhoGAP18B-PA regulates border cell migration.** (A) A diagram of border cell migration. Border cell cluster migrates through nurse cells during stage 9, and reaches the border of oocyte at stage 10. (B-E') Confocal micrograph of egg chambers at stage 10 are double labeled with DAPI (blue) for nuclei and with GFP (Green) to show the *slbo-Gal4* expression patterns. The arrows show the positions of the border cell clusters. The percentage scale represents the beginning or ending positions of border cell migration. (B, B') The control border cell cluster reaches the oocyte at stage 10. (C, C') The knockdown of PA prevents border cell migration. (D, D') The overexpression of PA impairs border cell migration. (E, E') The border cell cluster expressing both PA RNAi and PA reaches the oocyte. (F) Quantification of border cell migration. The x-axis denotes the percentage of stage 10 egg chambers for each genotype with each degree of migration. The extent of migration for all stage-10 egg chambers examined is measured as 0–5% (blue, no migration), 6–25% (yellow), 26–50% (pink), 51–75% (green), 76–100% (black, complete migration). The number of egg chambers examined for each genotype is given (n). Note that expression of PA efficiently rescues the migration defects caused by PA RNAi expression.

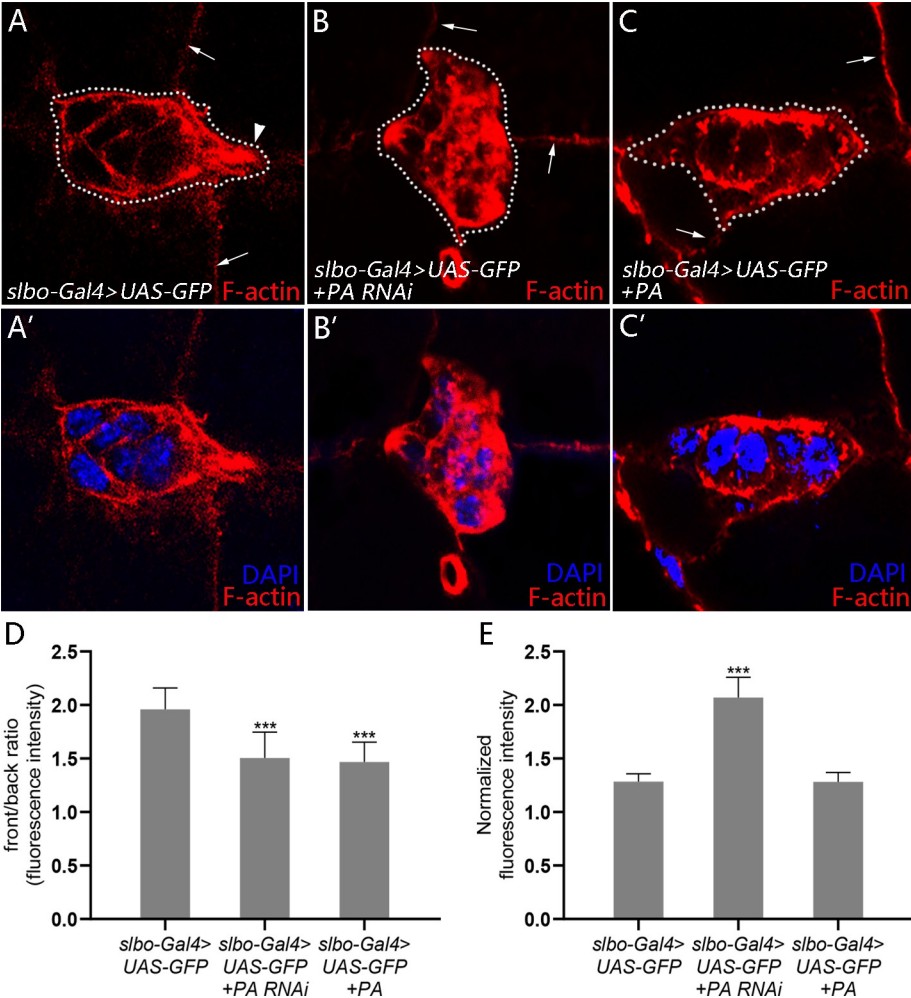

**Fig 3. RhoGAP18B-PA is required for F-actin organization in border cell cluster.** (A-C') Confocal micrographs of control, PA knockdown and PA overexpression egg chambers labeled with phalloidin (red) for F-actin and with DAPI (blue) for nuclei. Border cell clusters are marked by dotted lines and their movement direction is from the left to the right. F-actin protrusion is indicated by an arrowhead. Membrane F-actin staining between nurse cells is indicated by arrows, which is used for normalization of fluorescence in quantification of (D). (A, A') A *slbo>GFP* border cell cluster shows a polarized distribution of F-actin staining at the front. (B, B') A *slbo>GFP+PA RNAi* border cell cluster shows reduced polarity of F-actin distribution and elevated F-actin level. (C, C') A *slbo>GFP+PA* border cell cluster shows reduced polarity of F-actin distribution. (D) Quantification of F-actin staining front/back ratio of control, PA knockdown and PA overexpression border cell clusters (***P<0.001, t = 6.46, t test, n = 20; ***P<0.001, t = 8.08, t test, n = 20). See Materials and methods for details. (E) Quantification of normalized F-actin staining of control, PA knockdown and PA overexpression border cell clusters (***P<0.001, t = 17.26, t test, n = 20). See Materials and methods for details. Results are presented as mean ± s.e.m.

RhoGAP18B-PA was required for F-actin organization, especially for the polarized distribution of F-actin, during border cell migration.

## 3.4 RhoGAP18B-PA genetically interacts with Rac and LimK to promote border cell migration

We next sought to identify the Rho GTPase that RhoGAP18B-PA acted on. One candidate was the small GTPase Rac, which had been previously shown to promote actin polymerization and leading protrusion formation in border cells. We hypothesized that the migration defects and

abnormal F-actin in PA RNAi expressing border cells is due to the increased function of Rac. To test this hypothesis, we performed a series of genetic interaction experiments. We compared the migration degree of RhoGAP18B-PA RNAi lines to those carrying loss-of-function mutations in the three genes encoding *Rac* homologs (*Rac1*, *Rac2*, and *Mtl*) in addition to RhoGAP18B-PA silencing. Heterozygosity for the three mutations obviously suppressed the RhoGAP18B-PA silencing migration defect. Expressing a kinase inactive form of Lim Kinase (LimK.KI) also significantly suppressed the RhoGAP18B–PA silencing phenotype (Fig 4). These results suggested that RhoGAP18B-PA regulated border cell migration through Rac and

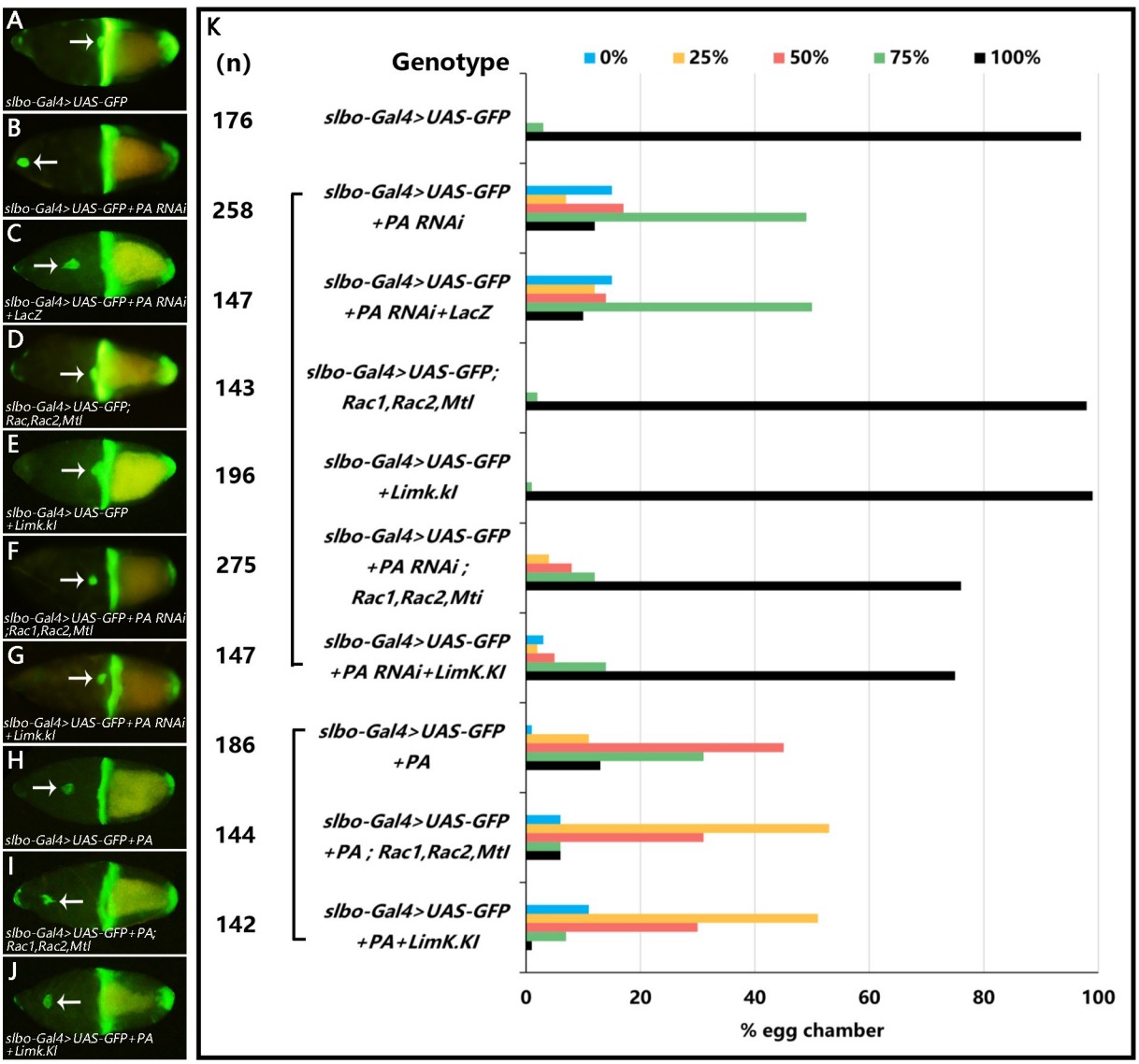

**Fig 4. Genetic rescue of migration defects in RhoGAP18B-PA knockdown in border cells by Rac mutant or LimK.KI.** (A-J) Stage 10 egg chambers of labeled genotypes show the *slbo-Gal4* expression patterns by GFP (Green). The arrows show the positions of the border cell clusters. (K) Quantitation of border cell migration. The x-axis denotes the percentage of stage 10 egg chambers for each genotype with each degree of migration. The extent of migration for all stage-10 egg chambers examined is measured as 0–5% (blue, no migration), 6–25% (yellow), 26–50% (pink), 51–75% (green), 76–100% (black, complete migration). The number of egg chambers examined for each genotype is given (n). Note that reducing the function of Rac or LimK can rescue the migration defects caused by PA RNAi expression and enhance the migration defects caused by PA overexpression.

LimK. Knockdown of RhoGAP18B-PA may elevate Rac function in border cells, so reducing Rac or LimK function could rescue migration defects of RhoGAP18B-PA knockdown. To test this hypothesis further, we compared the migration degree of PA overexpression lines to those carrying the Rac three mutant heterozygotes in addition to PA overexpression. The Rac three mutant heterozygotes obviously enhanced the migration defect of PA overexpression lines. Similarly, expressing of LimK.KI also enhanced the PA overexpression phenotype (Fig 4). Taken together, these results suggested that RhoGAP18B functions to reduce Rac function in border cells.

## 4. Discussion

In this report, we identified RhoGAP18B-PA, but not PC and PD, as a new regulator of border cell migration in the *Drosophila* ovary. RhoGAP18B-PA controlled the polarized distribution of F-actin and the levels of F-actin in the border cells. Moreover, reducing the function of Rac and LimK efficiently suppressed the migration delay of RhoGAP18B-PA knockdown border cells, and enhanced the migration delay of RhoGAP18B-PA overexpression. Thus, we propose a model in which RhoGAP18B-PA precisely regulates the levels of activated Rac, promotes polarized F-actin distribution in the entire border cell cluster, which in turn drives the migration of border cells (Fig 5). There are 27 RhoGEFs and 22 RhoGAPs in *Drosophila* (Flybase). Sprint, a RhoGEF, has been found to regulate border cell migration by controlling early steps of RTK endocytosis [21]. RhoGAP18B is the first RhoGAP found to be involved in regulating border cell migration. There may be additional RhoGEFs and RhoGAPs in *Drosophila* that regulate border cell migration. It should be pointed out that our conclusions are based on experiments used only one RNAi line. It is possible that there could be off-target effects. However, we think the possibility of off-target effects is low. The phenotypes of expressing PA RNAi fit the phenotypes of a RhoGAP gene knock-down. We also have done BLAST analysis with the target sequence of PA RNAi. The alignment scores of other candidate target genes are very low and no GAP genes are included.

Members of the Rho family of GTPase, which are comprised of Rho, Rac and Cdc42, play important roles in migratory cells and are well known in regulating actin organization [22]. These Rho GTPases show distinct functions in Swiss 3T3 cells. Active Rho induces actin stress fibers and focal adhesions, whereas active Rac facilitates lamellipodia protrusion formation, and active Cdc42 promotes filopodia protrusion formation [23, 24]. Other cell types also make

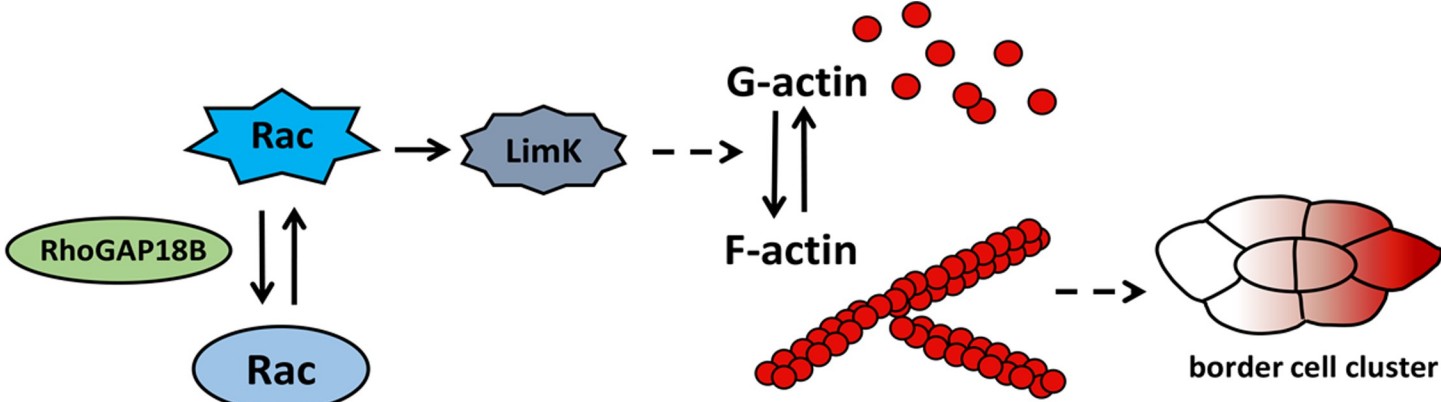

**Fig 5. A model showing the roles of RhoGAP18B-PA in regulating collective migration of border cells.** Rac function must be precisely controlled in border cell clusters, which is critical for border cells polarize to produce a major F-actin-enriched protrusion. Our data support that RhoGAP18B-PA regulates border cell migration by mediating Rac/LimK function.

different responses to the three small Rho GTPases. For example, in border cells, Rac function is critical for border cell migration. Expressing a dominant-negative form of Rac in border cells almost completely inhibits migration. However, expressing dominant-negative form of Rho or Cdc42 in border cells induces severe migration defects in less than 10% border cell clusters [19]. Therefore, although the possibility that RhoGAP18B may promote border cell migration through Rho or Cdc42 cannot be ruled out, we speculate that Rac is the major Rho GTPase that RhoGAP18B may act on during border cell migration.

Rac is also responsible for border cells to form leading protrusion. Focal activation of photoactivatable Rac in a single border cell results in the redirection of the entire border cell cluster [10]. The normal direction of border cell migration is induced by PVF signal secreted from the oocyte. Border cells use two receptor tyrosine kinase (RTKs), PVR and EGFR, to sense the guidance signal [5]. RTKs signal through Rac, Pak, LimK and Cofilin to regulate the actin organization and lamellipodia formation [19]. RTK activity demonstrates obvious front-back polarity in border cell cluster, with much more active RTK in the leading edge during the whole migration [25]. There are two models to explain Rac activity in border cells based on two different indicators of Rac activity. One is "Rac activity gradient" model. It is thought that RTK activity polarity would induce more Rac activity in the front of border cell cluster [10]. The other is "Two Rac pools" model. It revealed two switchable Rac pools at border cell supra-cellular cables and protrusions. The two pools integrate the direction and coordination of border cell migration [12]. Here, we find defined levels of RhoGAP18B-PA promote border cell migration. Both reduced and elevated RhoGAP18B-PA impairs border cell migration. Therefore, precise RhoGAP18B-PA amount is needed for border cell migration. We hypothesize that RhoGAP18B-PA may have space-specific protein localization in border cell cluster. Over-expressing PA RNAi or PA in border cell cluster may disturb the space-specific localization and subsequently distrubs space-specific Rac activity, and thus induce migration defect of border cells. Further study about the exact localization of RhoGAP18B will help us to understand the model for Rac activity in border cell clusters.

## Supporting information

**S1 Fig. The schematic diagram of UAS-PA plasmid.** The full-length PA coding sequence, with two EcoRI sites added in the 5' and 3' ends, was synthesized and subcloned to the pUAS-attB plasmid with EcoRI site. The DNA sequence synthesized for generating UAS-PA clone is:

```
gaattcATGGCCGGCGATACGGAGAACAAAAGGGGCTATCGCACCATATTTCGCAGCATAT
CCCAGGTGTTCTACGCCAATGCAAAAAACTCGAGTAATACGAACAGCAGCAGCAGTAACAA
CAACAATAATATCATCAACTACAACAACAACACCATCGAGAACAACAACGAGGGAATCGGC
CAACTAACAGTTACGCTCACCAGCATAACACCACCCACAGCCACATGGACGTCGACATCGC
CCACAATCTCCACGTGCACAACTGCGTCCGCATCGGCGGGATCATCCACCGCATCCTTGTC
GCCACTGGCTAGCAGATCAGCGCGATCGGCGGAGAATATATCCTGCAGCGAGTGTGGAAAC
CCAGATCCGTATCATGGCCAGGACATGGAGTTGGCAAAAGGTCAAGCCCTGACTCCCGAAG
TGCCAGCAGTGCCGGTTCCAGCTGCCAGTGCGGATCAAAACCAGAAGCTAAGGG
CCAGTTCCATGCTGGATCTGACCTCCAACCATCGCCATCAGAACGGCTTGAGGAT
TGTGCCAGTGACTGACATTCTGCGGGCGCAGCAGATCCAGGAGGACAATGAGGT
GCAGCAGACGCAGCAGTCAACGATTCTCCAGCGCTATAAGAGCGTCTCCATTGG
CAACCTGCTCGATCTGCCCGGCGAGGATGACAAGAAGTCCATCAAGAATCGCAT
GCGAATGAAGTTCGCCGTCAATAGCAATGTGTTTCGCATCAACCGTTCGCCGAAG
AGCGAGAAGAAGTCGCGTTCACGTCGAGGCCAGGTGAACAGTTTGTACCTGGAC
GAGCAGAAGTATCCGCTCTTTGCCGCTCCGCTCAGCGCACTGGAGCTCAACATG
ACCGATCATCCGAATGTTCCACGTTTCGTGGTCGACGTTTGCGCCTACATCGAAC
```

AGCCGGAGTGCATCGAGCAGGACGGACTGTACCGCGCATCGGGCAACAAGGTTCT
GGTGGACGAGCTGCGCAAGAAGCTGACGCACGTGTACGATCCCCGCTGGCTGAAG
ACGGACGATATCCACACGCTGACCAGTCTGCACAAGCAGTTCTTCCGAGAGCTCAC
CTCTCCGCTCATCACCCAGGAGGCCTACGAGCGACTGGGCAGGAGCCTTAACGAC
GACGCCGCCATCGAGCGGATGAGTTTGGCCTTTGACGACATGCCGGAGCCGAACC
GCTCCACGCTGCGCTTCCTCATTAGGCACTTGAC CAGAGTTGCC GCTGCCAGCG
CTTCGAATCCATGCCGTCCACCAATTTGG CCATCGTTTG GGGTCCTTGT CTGCTG
AGTG CCAATCAGAT ACAGCTGGACATTGGACGCA TGAACATGCT GGCCAAGGT
GCTGATCGAGA ACTATGATCG CATCTTTCATCCGGACAATG AGCGTCTAGT TTG
TTAGgaattc.
(DOCX)

**S2 Fig. Border cell migration degrees caused by RNAi or transgene expressions with**
***c306-Gal4.*** (A-G) Stage 10 egg chambers of labeled genotypes show the *c306-Gal4* expression
patterns by GFP (Green). The arrows show the positions of the border cell clusters. (H) Quan-
titation of border cell migration. The x-axis denotes the percentage of stage 10 egg chambers
for each genotype with each degree of migration. The extent of migration for all stage-10 egg
chambers examined is measured as 0–5% (blue, no migration), 6–25% (yellow), 26–50%
(pink), 51–75% (green), 76–100% (black, complete migration). The number of egg chambers
examined for each genotype is given (n). Expression of PA RNAi driven by *c306-Gal4* disrupts
border cell migration, but expression of PB or PC/PD RNAi did not affect border cell migra-
tion. Expression of PA also disrupts border cell migration and can rescue the migration defect
caused by PA RNAi expression.
(DOCX)

## Acknowledgments

We thank the Bloomington Drosophila Stock Center, the Vienna Drosophila RNAi Center,
the National Institute of Genetics Stock Center for their *Drosophila* stocks. We thank Core
Facility of Drosophila Resource and Technology, CEMCS, CAS for generating transgeic fly.
We thank Dandan Chu for editing this manuscript in English writing.

## Author Contributions

**Conceptualization:** Ping Wan.

**Data curation:** Fengyun Lei, Xiaoqing Xu, Jianhua Huang, Ping Wan.

**Funding acquisition:** Dan Su, Ping Wan.

**Investigation:** Dan Su, Ping Wan.

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
