## [Decision Letter · Decision Letter 0]

18 Nov 2022

PONE-D-22-28519Drosophila RhoGAP18B Regulates Actin Cytoskeleton during Border Cell MigrationPLOS ONE

Dear Dr. Wan,

Thank you for submitting your manuscript to PLOS ONE. After careful consideration, we feel that it has merit but does not fully meet PLOS ONE’s publication criteria as it currently stands. Therefore, we invite you to submit a revised version of the manuscript that addresses the points raised during the review process.

Please submit your revised manuscript within four weeks. If you will need more time than this to complete your revisions, please reply to this message or contact the journal office at plosone@plos.org. Please include the following items when submitting your revised manuscript:A rebuttal letter that responds to each point raised by the academic editor and reviewer(s). You should upload this letter as a separate file labeled 'Response to Reviewers'.A marked-up copy of your manuscript that highlights changes made to the original version. You should upload this as a separate file labeled 'Revised Manuscript with Track Changes'.An unmarked version of your revised paper without tracked changes. You should upload this as a separate file labeled 'Manuscript'.If applicable, we recommend that you deposit your laboratory protocols in protocols.io to enhance the reproducibility of your results. Protocols.io assigns your protocol its own identifier (DOI) so that it can be cited independently in the future. For instructions see: https://journals.plos.org/plosone/s/submission-guidelines#loc-laboratory-protocols. Additionally, PLOS ONE offers an option for publishing peer-reviewed Lab Protocol articles, which describe protocols hosted on protocols.io. Read more information on sharing protocols at https://plos.org/protocols?utm_medium=editorial-email&utm_source=authorletters&utm_campaign=protocols.

We look forward to receiving your revised manuscript.

Kind regards,

Carlos Oliva, PhD

Academic Editor

PLOS ONE

Journal Requirements:

Reviewers' comments:

Reviewer's Responses to Questions

**Comments to the Author**

1. Is the manuscript technically sound, and do the data support the conclusions?

Reviewer #1: Partly

Reviewer #2: Yes

2. Has the statistical analysis been performed appropriately and rigorously? 

Reviewer #1: Yes

Reviewer #2: Yes

3. Have the authors made all data underlying the findings in their manuscript fully available?

Reviewer #1: Yes

Reviewer #2: Yes

4. Is the manuscript presented in an intelligible fashion and written in standard English?

Reviewer #1: Yes

Reviewer #2: Yes

5. Review Comments to the Author

Reviewer #1: This manuscript shows a new required role for fly RhoGAP18B in cell migration in the ovarian border cells. Border cells are a good model system for investigating the genetic regulation of cell migration, as many genes required for this process also have roles in other migrating cells. RhoGAP18B is predicted to be an activating factor for the Rho family of GTPases, which include Rac, Rho, and Cdc42. Using RNA interference, the authors demonstrate that reduced levels of a certain RhoGAP18B isoform (PA) cell autonomously disrupts border cell migration to the oocyte. This is attributed to changes in F-actin organization, which is usually higher in the leading edge of the migratory cell group, but when RhoGAP18B is reduced, F-actin is more evenly localized. The defects are rescued by co-expression of RhoGAP18B-PA. Genetic interaction tests suggest that RhoGAP18B acts to de-activate Rac because reduction of Rac function suppresses the defects due to loss of RhoGAP18B. A connection is also shown to the downstream Lim Kinase. This work is original and improves the understanding of actin regulation in border cell migration, which may have broader implications.

While the results are generally clear and the data are sufficient to support the main points, more controls or a few additional experiments would enhance confidence in the conclusions. A limitation of the work is that the experiments used only one RNAi line to disrupt RhoGAP18B PA. Knockdown of the other isoforms do not have an effect, presumably because they are expressed at low levels in the ovary to begin with. The authors verify that the knockdown due to the RhoGAP18B PA line is specific to that isoform by quantifying RNA levels, and they show that the resultant defects can be rescued by co-expression of the wild type isoform. This suggests that the conclusions are likely to be correct, but it is possible that there could be off target effects, for example affecting a different RhoGAP that has a role in cell migration. The authors may want to test additional lines or at least acknowledge this possible caveat in the manuscript.

Minor concerns/suggestions:

- More details on how the authors measured F-actin levels in border cells versus nurse cells would be helpful. It is hard to see in the figure how the difference is only about 1.3-2.2 fold higher in border cells when it appears much brighter.

- I was unclear about the authors’ model for the spatial regulation of F-actin that occurs via RhoGAP18B PA. Normally F-actin levels are higher in the front of the cluster than in the back, but when RhoGAP18B is reduced by RNAi, the front/back levels are more even. The authors propose that RhoGAP18B acts to deactivate Rac primarily at the back of the cluster, which would be interesting to test in clonal analysis with different levels of RhoGAP18B PA. How do the authors propose this localized effect would be controlled? Maybe this issue can be discussed further in the text.

- In the discussion, the authors claim that RTKs are higher in the front of the migrating cell cluster and they cite Assaker (2010), but that citation shows that the activity, not the receptors themselves, are higher in the front. This should be clarified.

- Citations should include all authors unless the journal style indicates otherwise.

- In a few places, the paper needs minor editing for English usage, grammar, or typographical errors.

Reviewer #2: This manuscript is globally clear and well-written. The conclusion that RhoGAP18B controls border cell migration is supportive. But authors need to do some improvement for the publication at Plos One.

1. for results 1A, it is not clear for readers without Drosophila genetic background. Authors should clearly cite how they know RhoGAP18B has these isoforms. I guess this information of Fig. 1A is from flybase. Authors should clarify this clearly. In figure legends, authors should also mention clearly that RNAi is specific for each isoform, in case that readers will get confused if they are not Drosophila expert.

2. For the overexpression of PA, authors should check the effects on PA protein or mRNA. This is the necessary control experiment.

3. Some quantifications are missing in the figures, such as Fig. 1B.

4. qRT-PCR to detect the RNAi of different isoforms in which organ? whole body or the ovary? It is unambiguous. Please make it clearer.

5. Authors used heterozygous mutant of Rac1-3 gene to do rescue experiment. Normally heterozygous would not affect protein expression. It is unclear how this heterozygous mutant can reduce the increase of Rac1 activity (I guess that RhoGAP18B knockdown by RNAi can increase Rac1 active form). Can authors explain this unclear point?

6. Authors cited some references such as Rac1 by papers published before 2018? Since Rac1 have more novel findings in border cells, can authors put new recent studies in BC migration in their ref? Such as 2 recent papers: 1) Wang et al., iScience 2020; 2) Sijia Zhou, et al. Nat Commun. These two papers might explain the effect on F-actin network by RhoGAP18B-Rac1 when border cells lack prominent protrusions (and thus F-actin signals could be mainly from supracellular F-actin cables.

6. PLOS authors have the option to publish the peer review history of their article (what does this mean?). If published, this will include your full peer review and any attached files.

Reviewer #1: No

Reviewer #2: No

---

## [Author Response · Author response to Decision Letter 0]

9 Dec 2022

Reviewer #1: This manuscript shows a new required role for fly RhoGAP18B in cell migration in the ovarian border cells. Border cells are a good model system for investigating the genetic regulation of cell migration, as many genes required for this process also have roles in other migrating cells. RhoGAP18B is predicted to be an activating factor for the Rho family of GTPases, which include Rac, Rho, and Cdc42. Using RNA interference, the authors demonstrate that reduced levels of a certain RhoGAP18B isoform (PA) cell autonomously disrupts border cell migration to the oocyte. This is attributed to changes in F-actin organization, which is usually higher in the leading edge of the migratory cell group, but when RhoGAP18B is reduced, F-actin is more evenly localized. The defects are rescued by co-expression of RhoGAP18B-PA. Genetic interaction tests suggest that RhoGAP18B acts to de-activate Rac because reduction of Rac function suppresses the defects due to loss of RhoGAP18B. A connection is also shown to the downstream Lim Kinase. This work is original and improves the understanding of actin regulation in border cell migration, which may have broader implications.

While the results are generally clear and the data are sufficient to support the main points, more controls or a few additional experiments would enhance confidence in the conclusions. A limitation of the work is that the experiments used only one RNAi line to disrupt RhoGAP18B PA. Knockdown of the other isoforms do not have an effect, presumably because they are expressed at low levels in the ovary to begin with. The authors verify that the knockdown due to the RhoGAP18B PA line is specific to that isoform by quantifying RNA levels, and they show that the resultant defects can be rescued by co-expression of the wild type isoform. This suggests that the conclusions are likely to be correct, but it is possible that there could be off target effects, for example affecting a different RhoGAP that has a role in cell migration. The authors may want to test additional lines or at least acknowledge this possible caveat in the manuscript.

Using only one RNAi line is a limitation of our work, and we have acknowledged this limitation in the revised manuscript. We think the possibility of off-target effects is low. The phenotypes of expressing PA RNAi fit the phenotypes of a RhoGAP gene knock-down. We have done BLAST analysis with the target sequence of PA RNAi. The alignment scores of other candidate genes are very low and no GAP genes are included. 

Minor concerns/suggestions:

- More details on how the authors measured F-actin levels in border cells versus nurse cells would be helpful. It is hard to see in the figure how the difference is only about 1.3-2.2 fold higher in border cells when it appears much brighter.

The area of border cell cluster includes the area of nuclei, so the average fluorescence intensity is not high. The area of nurse cell only includes the membrane area. Therefore, the normalized fluorescence intensity of F-actin is not high as it appears. 

- I was unclear about the authors’ model for the spatial regulation of F-actin that occurs via RhoGAP18B PA. Normally F-actin levels are higher in the front of the cluster than in the back, but when RhoGAP18B is reduced by RNAi, the front/back levels are more even. The authors propose that RhoGAP18B acts to deactivate Rac primarily at the back of the cluster, which would be interesting to test in clonal analysis with different levels of RhoGAP18B PA. How do the authors propose this localized effect would be controlled? Maybe this issue can be discussed further in the text.

RhoGAP18B may be degraded by ubiquitination at the front. In fact, a recent study have proposed a “two Rac pools” model to replace the “Rac activity gradient” model （Zhou et al., 2022）. They used a new Rac probe, which is more feasible than previous probe to monitor subcellular Rac activity. It revealed two switchable Rac pools at border cell supracellular cables and protrusions. The two pools integrate the direction and coordination of border cell migration. Now, we are inclined to believe the “two Rac pools” model and we don’t think RhoGAP18B localized at the back of the cluster. We have discussed this point in the revised manuscript.

- In the discussion, the authors claim that RTKs are higher in the front of the migrating cell cluster and they cite Assaker (2010), but that citation shows that the activity, not the receptors themselves, are higher in the front. This should be clarified.

We have clarified this point in the revised manuscript.

- Citations should include all authors unless the journal style indicates otherwise.

We have included all authors in the citations.

- In a few places, the paper needs minor editing for English usage, grammar, or typographical errors.

We have edited some places of the paper for English writing.

Reviewer #2: This manuscript is globally clear and well-written. The conclusion that RhoGAP18B controls border cell migration is supportive. But authors need to do some improvement for the publication at Plos One.

1. for results 1A, it is not clear for readers without Drosophila genetic background. Authors should clearly cite how they know RhoGAP18B has these isoforms. I guess this information of Fig. 1A is from flybase. Authors should clarify this clearly. In figure legends, authors should also mention clearly that RNAi is specific for each isoform, in case that readers will get confused if they are not Drosophila expert.

We have clarified these points in the revised manuscript.

2. For the overexpression of PA, authors should check the effects on PA protein or mRNA. This is the necessary control experiment.

The pUAS-attB plasmid is broadly used in Drosophila experiments and works well. The generated UAS-PA plasmid has been sequenced. Generally, we do not measure the expression efficiency of pUAS-attB plasmid.

3 . Some quantifications are missing in the figures, such as Fig. 1B. 

Fig. 1B shows the different transcript compositions in adult fly body and ovary. There is no T test. So Fig. 1B is different from Fig. 1C.

4. qRT-PCR to detect the RNAi of different isoforms in which organ? whole body or the ovary? It is unambiguous. Please make it clearer.

Whole body. We have made it clearer in the revised manuscript.

5. Authors used heterozygous mutant of Rac1-3 gene to do rescue experiment. Normally heterozygous would not affect protein expression. It is unclear how this heterozygous mutant can reduce the increase of Rac1 activity (I guess that RhoGAP18B knockdown by RNAi can increase Rac1 active form). Can authors explain this unclear point?

Yes. Heterozygous mutant of Rac1-3 gene would not affect border cell migration, but it could rescue border migration defects caused by RhoGAP18B knockdown. This genetic interaction result suggests that RhoGAP18B knockdown by RNAi can increase Rac active form.

6. Authors cited some references such as Rac1 by papers published before 2018? Since Rac1 have more novel findings in border cells, can authors put new recent studies in BC migration in their ref? Such as 2 recent papers: 1) Wang et al., iScience 2020; 2) Sijia Zhou, et al. Nat Commun. These two papers might explain the effect on F-actin network by RhoGAP18B-Rac1 when border cells lack prominent protrusions (and thus F-actin signals could be mainly from supracellular F-actin cables.

We have added these two references and some introduction in the revised manuscript.

---

## [Decision Letter · Decision Letter 1]

2 Jan 2023

PONE-D-22-28519R1

Drosophila RhoGAP18B Regulates Actin Cytoskeleton during Border Cell Migration

PLOS ONE

Dear Dr. Wan,

Thank you for submitting your manuscript to PLOS ONE. Your revised manuscript has been read by the two original reviewers and both of them feel that the paper has improved and is suitable for publication in PLOS ONE, however,  one of them has recommended minor changes to correct language errors . Therefore, we invite you to submit a revised version of the manuscript that addresses the points raised.

We look forward to receiving your revised manuscript.

Kind regards,

Carlos Oliva, PhD

Academic Editor

PLOS ONE

Journal Requirements:

Reviewers' comments:

Reviewer's Responses to Questions

**Comments to the Author**

1. If the authors have adequately addressed your comments raised in a previous round of review and you feel that this manuscript is now acceptable for publication, you may indicate that here to bypass the “Comments to the Author” section, enter your conflict of interest statement in the “Confidential to Editor” section, and submit your "Accept" recommendation.

Reviewer #1: All comments have been addressed

Reviewer #2: All comments have been addressed

2. Is the manuscript technically sound, and do the data support the conclusions?

Reviewer #1: Yes

Reviewer #2: Yes

3. Has the statistical analysis been performed appropriately and rigorously? 

Reviewer #1: Yes

Reviewer #2: Yes

4. Have the authors made all data underlying the findings in their manuscript fully available?

Reviewer #1: Yes

Reviewer #2: Yes

5. Is the manuscript presented in an intelligible fashion and written in standard English?

Reviewer #1: No

Reviewer #2: Yes

6. Review Comments to the Author

Reviewer #1: The paper has improved in revision and is now clearer.

Here are two minor corrections on standard English language usage in response to question 5:

section 3.4 heading should say "genetically interacts" instead of "genetic interact"

figure 4 legend should say "defects in RhoGAP18B-PA knockdown in border cells" instead of "defects of PA knckdown border cells"

Reviewer #2: All my comments have been addressed. And recommend for the acceptance of this manuscript for publication.

7. PLOS authors have the option to publish the peer review history of their article (what does this mean?). If published, this will include your full peer review and any attached files.

Reviewer #1: No

Reviewer #2: No

---

## [Author Response · Author response to Decision Letter 1]

3 Jan 2023

Reviewer #1: The paper has improved in revision and is now clearer. Here are two minor corrections on standard English language usage in response to question 5: section 3.4 heading should say "genetically interacts" instead of "genetic interact" figure 4 legend should say "defects in RhoGAP18B-PA knockdown in border cells" instead of "defects of PA knckdown border cells".

We have corrected the two language errors in the revised manuscript. Thank you for your comments.

Reviewer #2: All my comments have been addressed. And recommend for the acceptance of this manuscript for publication.

Thank you for your comments.

---

## [Editor Report · Decision Letter 2]

5 Jan 2023

Drosophila RhoGAP18B Regulates Actin Cytoskeleton during Border Cell Migration

PONE-D-22-28519R2

Dear Dr. Ping Wan,

We’re pleased to inform you that your manuscript has been judged scientifically suitable for publication and will be formally accepted for publication once it meets all outstanding technical requirements.

Kind regards,

Carlos Oliva, PhD

Academic Editor

PLOS ONE

Additional Editor Comments (optional):

There are still few typos, that should be corrected

Page2, line 4

“Rac signals downstream to Lim kinase (LimK), which can meditate the activity”

Should be "mediate"

Page 4: **Results** should be preceded by the number 3.

Page 7, line9: “drosophila” should say *Drosophila* 

Page 7, third paragraph, line 5: it says “Pac” is it Pak?

---

## [Editor Report · Acceptance letter]

8 Jan 2023

PONE-D-22-28519R2 

*Drosophila* RhoGAP18B Regulates Actin Cytoskeleton during
Border Cell Migration 

Dear Dr. Wan:

I'm pleased to inform you that your manuscript has been deemed suitable for publication in PLOS ONE. Congratulations! Your manuscript is now with our production department. 

Kind regards, 

on behalf of

Dr. Carlos Oliva 

Academic Editor

PLOS ONE